# What do editors of medical journals think about opportunities and barriers to advancement in the publication of plain language summaries? A qualitative analysis

Karen Gainey *, Kirsten McCaffery, Danielle Muscat

Sydney Health Literacy Lab, School of Public Health, Faculty of Health and Medicine, University of Sydney, New South Wales, Australia

* email@karengainey.com

## Abstract

### Background

Plain language summaries (PLSs) are short summaries of research articles written for a diverse audience, using plain, easy-to-understand language. Although they have been adopted by many health and medical journals, we have a limited understanding of how they are viewed by journal editors. This remains an important gap because of the role journal editors play in the decision-making process for the publication, implementation, and dissemination of PLSs. To address this, the aims of this qualitative study were to: (a) gain a better understanding of the decision-making process that governs the publication of PLSs at the journal and publishing group levels; (b) explore attitudes and perspectives of journal editors towards PLSs; and (c) identify and explore barriers and facilitators to the implementation of PLSs by journals and journal publishers.

### Methods and findings

We conducted semi-structured interviews with 20 participants who represented 23 journals and eight publishers/publishing groups. Using reflexive thematic analysis, we developed five themes: (1) When good intentions clash with practical realities; (2) Whose job is it anyway?; (3) A cautiously optimistic approach to support from artificial intelligence (AI); (4) Blind spots and broken loops; (5) A 'One size fits all' approach doesn't work: the need for novelty. Discussions with participants highlighted the differing approaches taken by journals to prioritising PLSs, often governed by barriers such as resource allocation and the need for standardisation within publishing groups. While many participants showed initiative to overcome barriers, often in their own time, they noted challenges associated with a lack of PLS readership data and

**Data availability statement:** The datasets generated during and/or analysed during the current study are available from the Sydney eScholarship Repository at The University of Sydney, Australia and can be accessed via https://hdl.handle.net/2123/34296 DOI: 10.25910/mep6-2r15. This data is licenced under Creative Commons Attribution-Non-Commercial 4.0.

**Funding:** The author(s) received no specific funding for this work.

**Competing interests:** I have read the journal's policy and the authors of this manuscript have the following competing interests: Karen M. Gainey, Kirsten J. McCaffery and Danielle M. Muscat have completed the International Committee of Medical Journal Editors (ICMJE) uniform disclosure form at http://www.icmje.org/coi_disclosure.pdf and declare no support from any organisation for the submitted work; no financial relationships with any organisations that might have an interest in the submitted work in the previous 3 years; and no other relationships or activities that could appear to have influenced the submitted work, with the exception of Health Literacy Solutions Pty Ltd, at which Kirsten McCaffrey and Danielle Muscat are directors. Kirsten McCaffery and Danielle Muscat are Editors of health and medical journals. This does not alter our adherence to PLOS ONE policies on sharing data and materials.

role clarity in the PLS production and dissemination pipeline. Opinions were mixed on the integration of AI and the integration of alternate formats for PLSs.

## Conclusions

By embracing technology such as artificial intelligence, integrating innovative formats and exploring distribution channels including via consumer groups and social media, journal publishers can help elevate and expand the reach of PLSs.

## Introduction

Since the first medical journal was published in 1793, health and medical journals have evolved, introducing new features and formats [1]. One such feature is plain language summaries (PLSs), which are short summaries of research articles written for a diverse audience, using plain, easy-to-understand language [2]. The adoption of PLSs has steadily increased since they were first introduced in health and medical journals in the 1990s [3]. Interest in PLSs has also grown, with research focusing on varying aspects including PLS characteristics [4–6], the perceptions and preferences of PLS readers [7–10] and the effectiveness of varying formats [11–13]. Most commonly, PLSs are text-based; however, many journals offer alternative formats incorporating audio and visual elements, e.g., infographics [14]. A systematic review published in 2022 [15] found 90 articles that reported on criteria and outcomes of PLSs. Most articles reported recommendations on the use of language (e.g., avoid jargon and use passive voice), PLS structure (e.g., suggested length and the use of headings), and PLS content (e.g., guidance through "Who, What, Where, When, Why and How?" questions [15 p12]). However, study designs were too heterogeneous to identify definite criteria for high-quality PLSs, and few studies identified the effects of various PLS features on accessibility, understanding, knowledge, communication of research, and empowerment [15].

Research that has focused on the implementation of PLSs has been much more limited. We conducted a scoping review in 2023 [16] and found that very few journals (27/526, 5.1%) publish PLSs, with most of these being an optional inclusion. Guidelines for writing PLSs vary markedly in terms of detail and scope, which is a similar finding to that of other research studies [3,17]. When we assessed compliance with PLS guidelines, we found varying levels of compliance between published PLSs and the guidelines of the journals in which they were published, with areas for improvement in terms of the use of jargon, readability, and the content of the PLS [18]. Qualitative research with consumers also suggests that, even when implemented, PLSs are not always easy to find or access particularly in an increasingly crowded online health information landscape [7].

This emerging work suggests that while some aspects of PLSs have been well studied, we lack a fundamental understanding of other areas. The perspectives of editors of health and medical journals about PLSs have not been well explored. A review of the literature yielded only one study conducted by Open Pharma to

determine whether the perspectives of journal editors aligned with the Open Pharma recommendations for PLSs that were published in 2021 [19]. Since most (19/26, 73%) journals analysed in this study did not permit PLSs from authors and only seven journals had published PLSs previously, decision-making around, and barriers and facilitators to, their implementation was relatively unexplored [19]. This remains an important gap because of the role journal editors play in the decision-making process for the publication, implementation, and dissemination of PLSs. To address this, the aims of the study were to:

a. Gain a better understanding of the decision-making process that governs the publication of PLSs at the journal and publishing group levels.

b. Explore attitudes and perspectives of journal editors towards PLSs.

c. Identify and explore barriers and facilitators to the implementation of PLSs by journals and journal publishers.

## Methods

### Research design

To address our exploratory research questions, we adopted a phenomenological qualitative research methodology [20]. Phenomenology is an approach aimed at understanding the lived experience of others [21]. This study was approved by the low-risk ethics committee at the University of Sydney, Australia (approval number: HE001170).

### Inclusion criteria

We were interested in exploring the views of editors, associate editors and editorial staff from leading health and medical journals. We aimed to include at least one editor from each of the top five publishing groups, based on journal count [22]. These are Springer, Taylor and Francis, Elsevier, Wiley and SAGE. Our inclusion criteria were:

• Job title is Editor-in-chief, associate editor or editorial staff

• Works or volunteers for a journal that publishes health or medical research articles

• Works or volunteers for a journal that publishes text-based plain language summaries or similar, e.g., key messages, highlights

• Works or volunteers for a journal that publishes research articles in English

• Self-reported ability to participate in an English-language interview

To engage with editors from journals that publish in the area of health literacy and health communication (given the relevance to PLSs), we included editors from these journals even if they didn't publish PLSs.

### Recruitment

We selected participants using a combination of convenience, purposive and snowball sampling, recruiting from four sources. We began with a list of journals that we knew published PLSs from our previous study [16]. We added health and medical journals from the top five publishing groups listed above and those in the area of health literacy and health communication. From this list, we obtained the names and email of editors from publicly available information sources such as journal websites and public profiles and emailed them with information about – and an invitation to participate in – our project. If we did not receive a response, we sent a reminder email two weeks after the initial email.

Second, we distributed an email via the professional networks of the research team. These networks comprised professional and personal contacts of the research team, i.e., journal editors with whom members of the research team know. We

reached out to these editors in the same manner as all other potential study participants; using the recruitment email and two-week follow-up protocol. Third, we developed an advertisement for the study for use on social media websites such as LinkedIn or X (previously Twitter) and an advertisement for Cochrane Engage, which is a community of several thousand volunteers globally who share expertise and experience on projects or tasks. These tasks include peer review of Cochrane systematic reviews or article translation. Fourth, using snowball sampling, we asked participants for referrals after each interview. All potential participants obtained via referrals were contacted in the same manner as previously outlined.

Interested participants were directed to a survey hosted on Qualtrics to register interest in the study, view the participant information sheet and consent form, and complete demographic questions. For those participants that provided consent via the Qualtrics survey, the research team checked eligibility, and those eligible to participate in the study were sent an email to arrange an interview date and time. Recruitment was conducted between 17/10/2024 and 14/03/2025.

## Sample size

We used the concept of information power rather than saturation to determine our sample size [23]. Information power is based on the concept that the more information participants hold that is related to the research question(s), the smaller the sample size needed [23]. During data collection, we evaluated the sample size continuously, balancing criteria suggested by Malterud and colleagues [23]. See Table 1 for a description of our process for considering information power and sample size.

## Data collection

Interviews were conducted between November 2024 and March 2025. Interviews followed a semi-structured interview guide that was developed and iteratively revised by the research team, drawing on existing literature related to PLSs. See S1 Text. The scope of questions included topics related to:

• Decision-making for publishing PLSs

• The intended audience for PLSs

**Table 1. Considerations for determining sample size using information power.**

| Items that impact information power | Explanation | Application in the current study |
| --- | --- | --- |
| Study aim | A broad aim requires a larger sample size than a narrow aim. | Our study aims are quite narrow, with a discrete focus on the publication, implementation, and dissemination of PLSs in health and medical journals. |
| Sample specificity | The more specific the experiences & knowledge of participants in relation to the research question(s), the smaller the sample required. | We recruited editorial staff from journals that did and did not publish PLSs, as well as those working directly with publishing groups. Our sample specificity could be considered dense given that participants belonged to a specified target group while also exhibiting some variation within the experiences to be explored. |
| Established theory | When stronger and more established theories are used, smaller sample size is required. | This study is based on the theoretical principle of phenomenology, with the interviews accessing the perspectives of editorial staff based on their experiences with the decision-making process involved in publishing PLSs. |
| Quality of dialogue | When there is stronger dialogue in interviews, a smaller sample size is required. | This was the fourth study by the interviewer (KG) in the area of PLSs published by health and medical journals and the second qualitative study involving interviews, so the interviewer was experienced in the subject matter of the interviews and the process of qualitative interview technique. Coupled with the enthusiasm of participants to share their experiences, this yielded strong, high-quality interview dialogue. |
| Analysis strategy | Where a strategy is designed to provide an in-depth analysis of narratives, a smaller sample size is required. | Using reflexive thematic analysis, a thorough exploration of the data was conducted. |

- Guidelines for PLSs and compliance

- PLSs and peer review

- The use of tools involving artificial intelligence for producing PLSs

- Alternative ways of presenting PLSs

- The future of PLSs in journal publishing

We conducted 20 semi-structured interviews, guided by KG, who is a public health researcher trained in qualitative methodologies. During interviews, participants were encouraged to draw on any previous editorial experience to supplement that from their current position. Interviews were audio recorded and transcribed using the video conferencing tool Zoom. Transcripts were checked for accuracy by the first author and any identifying information was removed from interview transcripts. We offered all participants the opportunity to review a copy of the raw transcript to reflect on their responses with the interviewer, to offer additional insight or correct any misunderstanding. For the two participants (2/20; 10%) who chose this option, we emailed the raw transcript within two days of their interview. All comments or feedback made by participants were considered part of the transcript and included in the data analysis.

## Data analysis

We used reflexive thematic analysis to analyse interview transcripts [24–26]. Using this approach to qualitative research analysis, we identified and interpreted patterns or themes in the data [24–26]. Our approach involved the steps included in Table 2. Although we followed these steps in the order below, reflexive thematic analysis is not intended to be a linear process [24]. We moved between assigning codes to theme development several times as we interpreted the data at increasingly deeper levels. KG and DM were involved in data analysis.

## Reflexivity

Qualitative research is subjective, and the process of reflexivity enhances our understanding of this subjectivity [24–26]. By engaging in a process of reflexivity, we acknowledge that our analysis is naturally informed by our understanding and experiences and that the complete elimination of bias is not something that can be achieved in qualitative research and more importantly should not be an aim [25]. Engaging in a process of reflexivity, we acknowledge that our analysis is informed by our knowledge of PLSs, our belief in the importance of clear communication, and our experiences as both health consumers and editors. For this study, KG wrote analytic memos after each interview, which served as opportunities to reflect on what worked during the interview, what could be improved and any patterns or strong impressions from the interviews [26] and we held regular meetings to discuss findings and critically reflect on our positionality and

**Table 2. Steps in reflexive thematic analysis.**

| Step | Description |
|---|---|
| Familiarisation with the data | KG & DM became familiar with the data, noting key ideas and patterns. This was done by reading transcripts of the interviews and listening to audio files. |
| Data coding | KG & DM reviewed transcripts and tagged sections relevant to the research questions. |
| Initial theme generation | Using collaborative coding, KG & DM made connections within and between codes and developed a system for classifying the data over a series of discussions and meetings. |
| Theme development and review | KG & DM continued to review the data and develop themes in alignment with the research questions. |
| Refining, defining, and renaming themes | KG & DM summarised the key points of each theme, established a logical order in which to discuss the themes and identified titles for each theme. |

representation of research findings. In reporting the results, the study also aimed to centre participants' words descriptively to preserve their intention.

## Results

### Participant characteristics

Characteristics of participants included in this study are shown in Table 3. Most participants were male (11/20; 55%), resided in either the United Kingdom or Australia (13/20; 65%), had worked as an editor for at least five years (15/20; 75%) and reported making up to 10 editorial decisions each week (10/20; 50%). Four (20%) participants were unsure how many editorial decisions they made per week. Editors were most commonly affiliated with Taylor and Francis and Wiley publishing groups.

### Publishing characteristics

Given that some participants represented more than one journal, a total of 23 journals were represented by the 20 study participants. Most (N = 17) participants represented a journal and 5 represented a publisher/publishing groups. Almost half of the journals represented (N = 11) were affiliated with a professional society. Of the journals represented, half (N = 11) used the label 'plain language summary' and slightly more than half (N = 14) required a PLS with the manuscript, i.e., it was mandatory. Journal characteristics are presented in Table 4.

Through the process of reflexive thematic analysis, we developed five themes: (1) When good intentions clash with practical realities; (2) Whose job is it anyway?; (3) A cautiously optimistic approach to support from artificial intelligence (AI); (4) Blind spots and broken loops; (5) A 'One size fits all' approach doesn't work: the need for novelty. Across all themes, there were 15 sub-themes.

### Theme 1. When good intentions clash with practical realities

**1.1. Editors are purpose-driven to implement change.** Common to all interviews was an appreciation of the value of PLSs in supporting understandable health communication. While many participants expressed that they "*believed passionately*" or were "*very enthusiastic*" about PLSs as a form of communication to reach a broader audience, at minimum, other editors that we spoke to acknowledged the goal of PLSs "*to make sure all are served and everybody understands what is written in there*". Even the few participants associated with journals that did not produce PLSs spoke about some of the perceived benefits from their perspective, including expanding reach and readership:

*Participant 11: Having a broader readership, I think, can only help a journal. (Editor, Specialist journal).*

**1.2. Negotiating change.** In this context where participants placed value on PLSs, many reported 'taking it on themselves' to try to implement PLSs through their editorial roles, with some even having initiated the implementation of PLSs within their tenure.

*Participant 3: And so, it happened because the editors believed passionately it was really important, we took it on ourselves to do it. (Editor, General medical journal).*

*Participant 17: So, you know, we've been very enthusiastic about a number of different innovations, including…. obviously the plain language statements that you've seen. But that is all done by the editors negotiating that with [publisher X] to ensure that it can happen. If we didn't advocate for it, it wouldn't happen. (Editor, Specialist medical journal).*

**Table 3. Participant characteristics (N = 20).**

| Characteristic | n |
|---|---|
| **Age range (years)** | |
| 18-30 | 1 (5%) |
| 31-45 | 7 (35%) |
| 46-60 | 10 (50%) |
| 60+ | 2 (10%) |
| **Gender** | |
| Male | 11 (55%) |
| Female | 9 (45%) |
| **Country of residence** | |
| United Kingdom | 7 (35%) |
| Australia | 6 (30%) |
| United States of America | 4 (20%) |
| France | 1 (5%) |
| Germany | 1 (5%) |
| Russia | 1 (5%) |
| **Journal publisher [a]** | |
| Springer | 5 |
| Taylor & Francis | 4 |
| Elsevier | 5 |
| Wiley | 5 |
| SAGE | 5 |
| Other | 7 |
| **Job Title [b]** | |
| Editor-in-chief/Deputy Editor-in-Chief | 10 (50%) |
| Associate Editor/Senior Editor | 5 (25%) |
| Other [c] | 5 (25%) |
| **Years in current position** | |
| 1-3 | 5 (25%) |
| 3-5 | 6 (30%) |
| 5-10 | 6 (30%) |
| 10+ | 3 (15%) |
| **Approximate number of editorial decisions/week** | |
| <5 | 5 (25%) |
| 5-10 | 5 (25%) |
| 10-20 | 1 (5%) |
| 20-50 | 4 (20%) |
| 50+ | 0 (0%) |
| Varies | 1 (5%) |
| Unsure | 4 (20%) |

[a]Some participants represented more than one journal.

[b]Job title recorded by the participant in Qualtrics form.

[c]These job titles have not been included because their specificity could lead to identification of the participants.

**Table 4. Journal characteristics (N = 23).**

| Characteristic | n |
|---|---|
| **Journal publisher** | |
| Springer | 5 (22%) |
| Taylor & Francis | 1 (4%) |
| Elsevier | 5 (22%) |
| Wiley | 4 (17%) |
| SAGE | 2 (9%) |
| Other | 6 (26%) |
| **Journal publishes a PLS** | |
| Yes | 15 (65%) |
| No | 7 (30%) |
| Unsure [a] | 1 (4%) |
| **PLS is mandatory** | |
| Yes | 10 (44%) |
| No | 5 (22%) |
| Unsure [a] | 1 (4%) |
| NA (do not publish a PLS) | 7 (30%) |
| **Journal category** | |
| General medical | 6 |
| Specialist medical | 5 |
| Allied health | 3 |
| Public health, health promotion, healthcare policy | 4 |
| Health & medical education [b] | 5 |

[a]Participant could not recall this information and since the journal was archived, the author instructions were not available to reference.

[b]includes health literacy and health communication journals

When editors reflected on the process they went through to implement change, they reported varying levels of support from their journal's publisher. Some editors relayed how they had to negotiate with or "*convince*" their publisher of the value of PLSs. At times, such negotiation was centred around the need to demonstrate to the publisher that any change was financially justified.

*Participant 13: So, we'd have to convince the publishing company … you know, this is worth investing in, and their response will be 'Show me how it affects our bottom line'. (Editor, Health education journal).*

*Participant 17: For them, most of our requests have a cost, right?…But if they can see that it makes a difference to downloads, citations or the prominence of the journal, they're supportive. Okay? So, you know, it's, that's very, I mean, we have a good relationship. But it's a very transactional relationship. (Editor, Specialist medical journal).*

**1.3. Uniformity and autonomy.** For others, change was complicated by existing structures and processes established at the publisher level. Although many editors were open and supportive of approaches to encourage authors to submit PLSs, they had to work within the parameters of their journal and/or publishing group. A very tangible example of this was the constraints that pre-defined submission portals posed for PLS implementation, i.e., there not being a separate field for PLSs in the submission portal as there is for the manuscript abstract and other items. Publishers' preference for their

journals to have a distinct, unified style was also discussed in the context of PLS author instructions. Some editors felt that this prevented them from being able to offer PLSs in a way that best suits the journal's target audience.

Given such constraints, some participants that we spoke to felt that change needed to occur at the publisher level to "*harmonise the publishing practices of all journals, so they all fit best practice*" (Participant 6, Editor, General medical journal). Others, however, seemed to be more autonomous and were able to provide examples of times in which they were able to easily instigate change.

*Participant 1: We did want to introduce plain language summaries as a requirement in the author guidelines, and they were quite supportive of that. It wasn't a standard thing for them, at least not at that time. (Editor, Health education journal).*

Discussions about autonomy were often held in the context of broader reflections on the relationship between each editor/journal and their publisher, and also with regards to whether journals were owned by a publishing group or a medical society. Those journals affiliated with a society tended to have more autonomy than those owned by a publishing group. Editors noted how this autonomy enabled them to enact change more quickly and easily.

*Participant 8: My contract is with the Society to act as the editor-in-chief. What that means, though, is that I, via the Society, have had a fairly open license to shape the journal's editorial policies as I see fit. And that was true for the editors before me because this is not a publisher-owned editor, this is a society-owned journal. So, the highlights were something that I put in place when I became editor-in-chief. (Editor, Public health journal).*

*Participant 13: Kind of frustrated…with regard to the homogenisation of what publisher X is doing with their journals just so they have consistency in the way they look without being very supportive of the tailoring it, how we tailor it to our situations in terms of what our aims and scope are. Now, they give us some latitude to do that, but it's within the instructions to the authors which are just almost like suggestions….So we have a great deal of control over content, but we do not have much control over how things are structured in the manuscript. That's where they're one size fits all. And we don't have a lot of control over what budget we get. (Editor, Health education journal).*

A few of the participants were associated with journals that did not produce PLSs and shared their thoughts on why this was the case. One person suggested that the actions of the journal demonstrated that they saw no purpose in including PLSs in their journal because their only audience was other professionals, i.e., medical professionals and researchers not experts in the particular field covered by the journal. For another participant, the decision to not include PLSs was a lack of prioritisation. Both participants thought their respective journals were missing out on an opportunity to reach a broader audience by not including PLSs, especially when publishing open access.

*Participant 11: I think that in many of these journals, including mine, there are too many clinicians understandably, who are domain-specific. So, ours is a [X] journal. It's read by [Xists] or specialists in [X]. Some people can access it, but they're not the intended audience. So, you know what, this is not really an issue. It's not really a, there are other things that need to be discussed. There are other priorities, and so I think that's a little bit short-sighted…. So, you know, I'd like to. But I think that's what holds back our journal and other journals. It's simply not seen as a priority. (Editor, Specialist medical journal).*

*Participant 15: I don't think they [the publisher] see the journal, as even though it's open access as this public facing. I don't think they've moved, progressed to that level. And I get that vibe also from attending the conference that you're there to communicate your science with other scientists, and that's the audience that you're communicating with….they don't see a purpose of having a plain language summary, even though it's an open access journal, and so many other people are reading it. But yeah, I find that society quite insular. (Editor, General medical, Allied health & Health education journals).*

**1.4. Going "over and above" in the context of resource limitations.** Many editors reported operating with limited resources, leading to the need to prioritise projects and activities, with those related to the delivery of PLSs often receiving less attention. Participants often expressed that their time with PLSs was "*over and above*" what they considered their normal editorial role. This stood as a challenge to implementation and sustainability given existing constraints on time and resources.

*Participant 3: When I worked at [journal X], we had a lot of conversations about, do we have the time to do this? How are we going to do this? Who's going to do this? … And you know we ended up writing them you know, over and above what our normal job was. (Editor, General medical journal).*

*Participant 16: She's just doing that on her own, I think, like not from the Journal, and so on. So, you know, it's a resources question. Who's got the resources and the skill and the interest? (Editor, Public health journal).*

Conversations about resource limitations extended beyond human resources. Many participants reflected on the "*shoestring*" budgets allocated to their journals and discussed how they needed to weigh decisions about PLSs in such contexts. There were clear opportunity costs to generating PLSs that editors considered in their daily decision-making.

*Participant 3: I suspect that, for example, a lot of specialty journals which are run on a shoestring don't have the capacity to even generate plain language summaries. (Editor, General medical journal).*

*Participant 2: With regards to the resources we have and the benefits we might get out of those resources… that is not a priority. (Editor, Specialist medical journal).*

Given the multiple tasks editors had to juggle, once put in place, processes involving PLSs such as author instructions and peer review, were rarely reviewed.

*Participant 3: Are the headings that we have there the right ones? I don't know. But you know, we've got a whole, I've got a whole process of things that I'm doing at the moment, and this serves a function. So, we've decided to leave them as they are at the moment. (Editor, General medical journal).*

**Theme 2. Whose job is it anyway?**

Interviews touched on the different people involved in the publication pipeline, including authors, editors, peer reviewers, and consumers and their role (or lack thereof) in PLS development. A lack of clarity and ownership in terms of PLS production was thought to contribute to the variable quality of PLSs.

**2.1. Authors.** Participants in our interviews generally thought that authors were not proactive about including PLSs in their research articles or developing them to a high standard. This was attributed both to a lack of familiarity with PLSs or plain language best practices, as well as a lack of priority. Complicating this was the observation that many authors do not seem to read the author instructions for the journal, either submitting a PLS that isn't suitable or not submitting one at all.

*Participant 1: Because in reality, people don't read the author guidelines. So that's why we often get the 1st versions without having a plain language summary at all…. Quite often they will just sort of do it as a bit of an afterthought, I think, and they'll just do it to get through the requirements, but they're not really thinking about 'Is this understandable for an average person'? And some of them, I guess that's not really familiar to them… (Editor, Health education journal).*

While there was an acknowledgement that *"there are some topics…that are extremely difficult to put into PLSs like … survival curve, extrapolation, mathematics in multi microsimulation models…",* the general sentiment was that, irrespective of the topic, authors are often tempted just write the PLS with minimal effort so the manuscript can be published. Reflecting on the submission process, participants noted that PLSs are often the last part of the manuscript that authors write and are often a direct copy of other sections such as the abstract or conclusion, with little effort made to consider the audience.

*Participant 2: The authors not (sic) necessarily understand what is meant with the patient summary. So, we very often just get a copy-paste from the conclusions of the abstract, which obviously is not the aim behind it. And I think there's a clear problem with the understanding of a scientist of what might be plain language. (Editor, Specialist medical journal).*

*Participant 4: They've done the work…they get to the end of the review, and they think, 'Oh, I've got to do this now.' And it is hard. It's a skill … actually being able to write in plain language is really difficult. (Editor, General medical journal).*

Participants in this study reflected on the varying levels of enthusiasm for PLSs on the part of authors. While some felt that authors most likely do not enjoy writing PLSs given that it's *"just such a painful thing…it's the thing that you hate the most",* other editors recognised that not all authors find writing PLSs a difficult or unpleasant task. Of the participants that we spoke to, people felt that the distinction could be attributed to the value authors attributed to PLSs as a way of communicating their research to more people.

*Participant 20: It's hard to put into words. But when a group of authors does the PLS because they want to, it's a different quality of PLS than when they're forced to….if it's written from like that good spirit like, we want to make this research understandable. (Publisher)*

Likewise, editors commented on the notable difference in the quality of PLS written by professional medical writers. Due to their scope and audience, some journals attract more articles from authors that have industry funding with the budget for a professional medical writer to produce a high-quality PLS. Editors commented that they rarely suggest changes to these PLSs. However, one editor pointed out that authors might be the best people to write the PLS because they are most familiar with the research and potentially have the most to gain.

*Participant 10: Papers that are sponsored by Pharma companies, they'll go through lots of levels of approval before they come to me. They'll be checked and double-checked, and you know they'll work with medical writers all these kind of things. So, things tend to be reasonably polished by the time they reach me. (Editor, Public health & Health education journals).*

*Participant 16: I do see the appeal of author written plain language summaries because the author's invested…The author knows the subject best, and is not likely to misstate, or less likely to… But the trade-off is the one we've discussed, which is, they also don't know how to talk about it. (Editor, Public health journal).*

**2.2. Editors.** In terms of editors' roles, perceptions were similarly varied. Some editors and members of their team took a proactive approach, intending to work with authors to improve PLSs. However, it was up to the individual how much time and effort they put into this area.

*Participant 8: I definitely have associate editors who are pretty good about going in and saying, 'Yeah, this doesn't make any sense. You need to clarify this, to authors, and there are others who don't necessarily pay that much attention to it. (Editor, Public health journal).*

 

*Participant 2: With regards of the resources we have and the benefits we might get out of these resources so, for me also, that is not a priority. But I would not necessarily neither pay too much attention on what the content is if that could not be improved. And if that indeed is plain language, or if that is still like too scientific to be called plain language. (Editor, Specialist medical journal).*

One participant noted that their journal had a team of "*structural editors*" who have an active role in PLS editing, but this was the exception rather than the norm amongst the participants we spoke with.

*Participant 3: So, we have a team of structural editors, and they find that they nearly always have to rewrite them quite substantially, because the authors, you know, they provide us with one, but it's often, you know, not very well written, or is kind of is too technical. So, we, one of our team will rewrite it into a form that we think is kind of the most accessible. (Editor, General medical journal).*

**2.3. Peer reviewers.** PLSs are rarely considered a notable part of peer review, nor a priority for peer reviewers. The participants that we spoke to felt that this is primarily due to a lack of formal processes to flag PLSs during peer review; PLSs are rarely included in peer review guidelines and reviewers are not prompted to comment on them specifically. When peer reviewers did comment on the PLS, most often it was about the content, rather than the language. For example, reviewers noted that PLSs might contain exaggerated or out-of-context findings rather than too much jargon, acronyms, abbreviations or complex language.

*Participant 3: Not really because it's not really a question that we're asking the reviewers. (Editor, General medical journal).*

Perceptions of peer-reviewers role in the implementation of PLSs varied between participants, with some agreeing that PLSs should be given more attention and others believing that is not the function of peer review given that the "*the most important target or objective we have for this is* [a review of] *scientific content methods*".

*Participant 2: I completely understand that they would not spend too much energy on what the patient summary contains. And that is yeah, fine. (Editor, Specialist medical journal).*

**2.4. Consumers.** Some editors expressed support for consumer involvement in PLS co-design, acknowledging that it is "*a good thing is that patients get more and more involved*" and that "*some will take a huge benefit out of it*". However, they noted that it was not for everyone. An additional common concern we heard from editors was that including PLSs could slow down the publication process. This was expressed by Participant 2 about the involvement of consumers in the production of PLSs because they might lack an understanding of the research process.

*Participant 2: What would be helpful indeed, like having a patient or a lay re-reading this part before publication. But then, I'm not really sure we would be able to find somebody who would provide that in a fruitful fashion. Because, yeah, either you're familiar with the scientific publication process...or you're not. And then you might probably slow down the entire publication process. (Editor, Specialist journal).*

There was an acknowledgement that co-design with consumers needed to be done meaningfully or not at all. A few journals or publishing groups achieved this with the establishment of consumer advisory panels. At least one member of the panel would review all PLSs and provide feedback according to guidelines for consumer reviewers. There was, however, disagreement about how to allocate PLSs to consumers. Some editors felt that ideally, consumers should

review summaries that are on topics in their area of expertise or for which they have lived experience. Others deliberately matched consumers with summaries outside their experience, believing that consumers are more effective reviewers when they are not too familiar with the subject matter.

*Participant 3: It needs to be 2 or 3 people, so they don't feel like they're just the token person in a group who's there just to tick a box. Then you have to have a strategy for how you involve them and all of that kind of thing… I think that I don't want to do it half-heartedly…So I think it's around not just sort of tacking them onto your kind of current processes and expecting them to get involved. But actually, you know, really meaningfully involving them in all the relevant parts of the processes. (Editor, General medical journal).*

**Theme 3. A cautiously optimistic approach to support from artificial intelligence (AI)**

Having noted that PLSs take time to write effectively, and the skill level of authors will vary, many participants expressed enthusiasm for the potential of AI technology to generate PLSs. Here, AI was seen as an opportunity for authors who either lack the skill, time or interest to write a PLS to be able to produce one of better quality. Although believing AI is a useful tool, all recognise the need for a 'human in the loop' to check the AI output as the accuracy varies.

**3.1. A good starting point.** There was widespread acknowledgment that PLSs require skills that are different to those needed for academic writing. This is where participants saw AI as being of most benefit to authors, reflecting on their challenges in writing PLSs. Helping authors make a start with a PLS was seen as a useful way of using AI, "*…even if it produced a 1st draft*". Generating infographics was seen as another common advantage of AI. With the number of journals offering visual abstracts, editors were in favour of authors or journals themselves potentially using AI to produce high-quality infographics as another way of communicating with a broader audience.

*Participant 3: I mean, I know people who are using AI. I know authors have told me they're using AI to generate info-graphics of their content. And again, you know, given how hard it is to, you know, develop nice infographics, I could see that that's actually possibly quite a useful tool, use of it. (Editor, General medical journal).*

**3.2. The responsible use of AI.** While enthusiastic, nearly all editors recognised that there would always need to be a 'human in the loop' to check the output as the accuracy of AI varies. They felt that evidence providers have a responsibility to ensure that information in PLSs is accurate and something readers can rely on. Some participants also felt that authors would need to declare the use of AI in the production of PLSs, as they would for any other part of their manuscript. Most journals have a policy for the reporting of AI; however, it does not specify whether this applies to the main manuscript or all parts of the submission. Most participants agreed that AI use is in its infancy and AI policies will become more detailed in the future.

*Participant 4: Yeah, yeah, I think we're a long way off, a long way off. I wouldn't want to send something out without having checked very, very thoroughly that it was, you know, actually properly representing the review. (Editor, General medical journal).*

*Participant 3: I mean, obviously, there's always gonna have to be human intervention in it. But I don't think it's the kind of thing where it would be wrong to explore using it. (Editor, General medical journal).*

**3.3. Does AI pose unforeseen risks?.** A few participants commented on their concern about the introduction of AI to write PLSs before the consequences of doing so are fully understood by those in academia and publishing, including risks associated with putting data in an open system and how this might impact copyright. One participant pointed out that, like

humans, AI systems have biases. This reinforces the notion in sub-theme 3.1 that all AI outputs should be checked by a human for accuracy and to check for any biases in the data they provide.

> *Participant 19: …which I think is a challenge, kind of if an author doesn't understand the consequences potentially of that, like putting their data in kind of an open AI system. But that's the same for anything, not just scientific data. So, there's definitely risks associated, and I think it will come with, again education of authors and them understanding how it works. (Publisher).*

> *Participant 17: Here's the rate-limiting step. Right, for you to use a large language model either to generate a plain language summary, or to generate a figure you have to upload the paper to chat GPT, or whatever machine AI machine that you want to use, and because our most of our papers are copyright, and that copyright is held by publisher X, they're not free, they're not free to access. Whereas if you put it out there on the Internet, in Chat GPT, it's free for everybody. And the license, and my understanding is the copyright, once you've uploaded it is held by the large language model. (Editor, Specialist medical journal).*

> *Participant 3: The one thing that I would say, obviously, we just need to be really careful about with AI is, we know, all these biases that we have. There are within the training programs for AI. We know that they, you know, they under-represent, you know, certain populations. They under-represent certain parts of the population as well. So, I think that using AI in a way that is kind of mindful is probably fine. But we do need to understand what its biases are. (Editor, General medical journal).*

**Theme 4. Blind spots and broken loops**

Editors told us that journals rarely receive feedback on the quality or usefulness of the PLSs they produce. This is because there is no process for this, and unsolicited feedback is rarely provided. This makes it difficult to know if they are reaching their target audience and whether there are ways in which they could deliver a more effective product. This is complicated by the broad nature of PLS readers and the varied audiences that journals strive to cater for with their PLSs.

   **4.1. Who is the actual target audience?.** Editors had varying opinions about the target audience for their journal's PLSs. For some journals, this is based on the scope and topics covered by the journal. Although most editors agreed that their PLSs were aimed at a non-expert audience, views differed in the definition of 'non-expert'. Some editors understood this as being health practitioners, researchers from other fields, policy makers and journalists. In these instances, most editors did not think that the PLSs in their journals would be of interest to a general audience, i.e., the general public. This is mainly the case where the scope of the journal was narrow, e.g., catering to a specific type or category of medical condition. Other editors considered anyone a potential reader of their journals' PLSs.

> *Participant 5: The general public, that will include all the peoples, like even the clinicians, the patients like the others, health researchers who can understand the maybe the statistical terminologies or the research terminologies. So, if you say the general audience, general readers, or general public, so that's fine, maybe. (Editor, Specialist medical journal).*

> *Participant 11: One of the themes that came up with our discussion of plain, of plain language summaries was the need for some doctors from different disciplines to be able to understand the material as well. (Editor, Specialist medical journal).*

   The lack of clarity around the target audience at times made it difficult to decide on the appropriate label for PLSs, with participants reporting *"many endless discussions about what we should call this".* While most journals use the label 'plain language summary' to convey the purpose of the summary, others opted for labels that were more audience-oriented such as 'Patient summary'.

*Participant 2: That's why we intentionally, indeed, selected the term patient summary, because it really targets the patient....to make sure that they understand that this part is for them, and that they might get some plain language information. (Editor, Specialist medical journal).*

**4.2. Evaluation is difficult without readership data.** Challenges pinpointing the audience of PLSs were further complicated by a reported lack of data on the actual readership of PLSs. Here, some participants acknowledged that there might be a difference between the target and the actual audience. This is complicated by the fact that most PLSs are attached to articles published open access. Unlike the subscription model, data on readership cannot be obtained from open-access journals. Editors we spoke with wanted to evaluate aspects of their journals' PLSs but were reluctant to do so without the evidence.

*Participant 12: Ultimately, our journals that publish these are open access, or if they are in a hybrid journal, then the plain language summary is always open access, but this makes it really difficult to track who is specifically reading them. (Publisher).*

*Participant 4: I would love to have some more data on who is using them and how they're using them properly. And I would really love to be able to revisit the template and the guidance and say, 'Well, hang on! Is this working? Do you like this?', you know, because I think I think there are things that could be tweaked, and I would like to do that, but not without the evidence. (Editor, General medical journal).*

**Theme 5. A 'one size fits all' approach doesn't work: the need for novelty**

Participants had many ideas about creative strategies to make PLSs appealing and accessible to a more diverse audience. These strategies included the introduction of novel formats beyond text-based PLSs and the use of novel avenues to distribute PLSs.

**5.1. Novel formats.** Despite challenges pinpointing the audience of PLSs and a lack of data on actual readership, there was a general acknowledgement that novel approaches to PLSs are needed, particularly to improve accessibility for diverse audiences. Several participants had experience working with plain language summaries of publications (PLSPs), across more than one publishing group. Although a minority in our sample, these participants were enthusiastic about this as a more novel format for PLSs. While PLSPs serve a similar function to that of PLSs, there are notable differences that make them more discoverable and freely accessible as they are not housed behind paywalls [14]. Also, at up to eight pages, they are considerably longer than PLSs, enabling more content to be included for the reader [14]. Some of the notable aspects of PLSPs participants pointed out were that they are a separate article category with their own DOI number, and as such, could be indexed by catalogues such as PubMed, Medline and Google Scholar. As they noted, PLSPs tend to have multiple authors, including at least one person from the authorship team of the original manuscript, along with patients/carers or someone with relevant lived experience. Of note, PLSPs tend to be peer-reviewed by patients or consumers.

*Participant 19: Yeah, I think it's an article type that's here to stay. Now, there's, I think, for publishers that do this type of content already, and the more the merrier, really, because it's the whole purpose of it is to make research more accessible to people that need it. (Publisher).*

*Participant 18: So, they (patients or consumers) generally peer review them just to make sure that the article sort of reflects the original publication. You know that they've not sort of cherry-picked any of the data or misrepresented any of the data, and that actually, it is written in the, in plain language in a way that they can understand… patient reviewers*

*also have a lot of experience in reviewing these types of articles. They know what they should look like. (Editor, Public health journal).*

More commonly, editors viewed the provision of accessible research content in formats that were not text-based as a positive step and one that might increase the journal's readership. This discussion often centred around readership diversity and concerns that many people may struggle with text-based information written in English, recognising *"it's a huge barrier that is in the way science is disseminated".* This included, for example, people from a non-English speaking (NES) background and those with chronic medical conditions that can impact visual or auditory processing and cognitive functioning. The formats discussed were podcasts, infographics and video/visual summaries or abstracts.

*Participant 10: There's lots of different ways to think about what a PLS is. So, at the moment, I think we're mostly focusing on the text. And then this is the more infographic-type side of things. But I think there's a lot of potential in all sorts of different formats, whether that's audio or video, or, you know, more interactivity. (Editor, Public health & Health education journals).*

*Participant 6: We have a podcast…we are in Apple podcast, Spotify… And we're doing some interviews and so on. So, I think that nowadays there's a huge uptake in podcasts for information as well, because people listen to it while [sic] commuted to work while cleaning their house. (Editor, General medical journal).*

However, there was an acknowledgement – once again – that different formats require additional and specific resourcing to be successfully implemented and maintained at scale across a journal or publishing group.

*Participant 3: I would really like us to move to even having video summaries…it takes time to do them, and this is one of the things I learnt when I was at [journal X]. We had three very experienced editors, and we thought, how hard can these be to write? They're really hard to write. I estimate it took us; it could take anything from 3 to 5 hours to write one…they're not trivial to do. (Editor, General medical journal).*

**5.2. Novel dissemination routes.** The availability of articles and their PLSs varied between journals, depending on the publishing model, i.e., subscription, open access or hybrid. Most participants agreed that journals publishing open access should include PLSs as this aligns with the principle of making research something that is widely available. There was no clear consensus about how best to deal with articles that are not published open access. The main issue raised by participants is that when articles are behind a paywall, so is the PLS. Members of the public and other potential readers are unable to access them as they are not likely to have a subscription, nor would they be willing to pay a one-off fee. Housing the PLS in a separate part of the journal website that is searchable and accessible was not seen as a solution because people might want to read the article as well as the PLS.

*Participant 2: Would be interesting to see how they could access it, and probably make sense to have, like the patient summary available outside of the open access or not open access wall. But then, I have doubts that somebody would be just happy with the let's say, five sentences. Probably they would like to look more into detail and get more information than just with this plain language summary, which I think is just a teaser and not necessarily the only information that should be getting out of an article. That is just too short, and I think not really helpful, I would expect. (Editor, Specialist medical journal).*

Due to the popularity of health information on social media, we asked participants about their experience integrating social media and PLSs and whether this was initiated by the author, or journal/publisher. Most stated that authors rarely

seemed to share PLSs of their work on social media, although some noted that they didn't pay close attention. One person explained that this might reflect the article's funding source rather than the author's choice. A few participants pointed out that many PLSs and PLSPs in particular are written by medical writers, for articles with industry funding. In such cases, social media is not the preferred method of promoting an article PLS or PLSP.

> *Participant 3: One of the things that we have considered is like, how do we distribute these, the articles on social media. For example, you could imagine that perhaps a box with the summary in it might be quite useful, a little social media tile, but don't do that routinely at this point. (Editor, General medical journal).*

> *Participant 12: There is a concern that people might see it as promotional if people were sharing it sort of far and widely, you know, so there may be a reluctance from authors to share it, or from the company sponsors to share it. (Publisher).*

One participant mentioned using a mixed approach of new and traditional media, i.e., newspaper, radio and television, as avenues for spreading health information in the form of PLSs. This was seen as a way of reaching even more people.

> *Participant 2: I think we would also need to think about having other medias taking up the information like if you would not only have your information inside of a scientific journal but maybe also like a normal newspaper or the Internet. I think that is where it becomes very helpful to get in this direction…the people being inside of a New York Times or whatever that they would be able to capture that information being let's say, attracted by that information because they might not necessarily be super scientific neither. But they might capture it and see, okay, that's interesting. And why not having an article or a paragraph on this information? And if you have indeed understandable information coming from our journal outside of our journal, then this might be something that you can achieve. (Editor, Specialist medical journal).*

Three participants mentioned that they disseminate PLSs or PLSPs through their growing network of consumer support groups. Those that used this approach, kept a spreadsheet of consumer groups and would match the topic of a new article to that of a consumer group and send them the associated PLS or PLSP. If they did not have any groups in their database that matched, the team would conduct an online search and add relevant groups, growing their potential reach.

> *Participant 19: It is a kind of more general method of accessing the content. And we know, we obviously share them with patient groups when they're published. So, it is going directly to groups to share if they want to. And patient groups, we know, have included them in their newsletters and in their own social media, and we publish on our social media, as well. (Publisher).*

> *Participant 12: What we do is we tend to make patient advocacy groups aware of them. And then they can choose to share it with their membership or not. (Publisher).*

> *Participant 10: Try where relevant to contact any relevant patient advocacy organisations. So, if they want to share content, then they can…it's letting people, the right people know it's there. (Editor, Public health & health education journals).*

## Discussion

This qualitative study with 20 journal editors representing 23 journals and 8 publishing companies provided valuable insight into the perspectives of journal editors about the publication of PLSs. While most editors were supportive of, or invested in, publishing PLSs, practical barriers to their implementation were consistently reported. These included barriers

related to resourcing, organisational commitment from publishers, difficulties assessing PLS readership and reach, and a lack of clearly defined roles, commitment and relevant skills among authors, peer-reviewers and editors. The future of PLSs was considered important in terms of adapting to emerging technologies such as AI, making use of innovative formats for PLSs to cater for a more diverse audience, and acknowledging some unexplored distribution channels including via consumer groups and social media.

The results of our study support and extend that of others related to PLSs which have largely ignored issues of implementation. Most similarly, Baróniková et al. [19] conducted a largely quantitative study of journal editors. While in common with our study, they identified implementation barriers including infrastructure challenges such as time, money and personnel, with mixed support for PLSs in peer review, there were some important differences in results. For example, in Baróniková et al. [19], 26% of journals stated that PLSs were not offered because they were not relevant to journal content, whereas we rarely heard this sentiment in our interviews. Rather, the main reasons we were given for journals not offering PLSs were because the target audience was primarily medical professionals or that the decision was made to focus on making the manuscript abstract more accessible on the basis that many people do not read past the journal abstract. These noted differences between our study and Baróniková et al. [19] could be due to the differing scope of journals in each study and the different data collection methodologies used in each study. Baróniková et al. [19] conducted a largely quantitative survey distributed electronically and our study comprised one-on-one semi-structured qualitative interviews. Both our study and Baróniková et al. [19] found that the target audience for PLSs was mixed, being mostly patients and patient groups, medical practitioners and researchers. Baróniková et al. [19] noted students as a common target audience, of PLSs, however, few participants in our study did so. Support for PLS in peer review was mixed in Baróniková et al. [19] as it was in our study.

This in-depth qualitative study provides novel insights into the decision-making process that govern the publication of PLSs, covering barriers, facilitators and attitudes of journal editors towards PLSs. The sample included a diverse voice within health and medical journal publishing, including both journal editors and those representing publishers or publishing groups. While we purposively engaged editors from journals that publish in the area of health literacy and health communication (given the relevance to PLSs), we did not purposively sample to include journals which publicise themselves as patient experience or participatory science journals in the same way. Also, we intentionally included editors associated with journals that did not publish PLSs and those for which PLSs were not mandatory. This provided us with rich interview data to more fully answer the study aims and was supported by rigorous and reflexive qualitative analysis methods. However, despite international recruitment, we did not have participants from journals based in Asia, South America or Africa. Coupled with the exclusion criteria of editors who did not speak English, we do not know if there are substantial differences in the perspectives, barriers or facilitators to the publication of PLSs in those geographical regions not represented.

The qualitative findings from this study point to several directions for future research and practice. Firstly, there is a need for practical support to generate PLSs for more articles. A lack of resources such as time, money and personnel are faced by many editors and publishers, posing pragmatic limitations to the publication of PLSs. Based on discussions with participants, any solution is likely to need publisher approval, however, it may be worth investigating the degree to which some of the processes involved might be automated. However, our findings point to the potential of generative AI tools to develop the first draft of a PLS and relatively minor modifications to journal submission portals to better accommodate the inclusion and review of PLSs. Structured support by way of communication and monitoring systems is needed to make the production and review of PLSs as easy and streamlined as possible. On a practical level, academic institutions could assist authors from all specialties of health and medicine by focusing on health literacy knowledge and plain-language skills via training. Open-access publishing has made it easier for anyone to access scientific research. However, it has made it more difficult for journals to assess their impact and in particular, the impact of the PLSs they publish. Future research should prioritise novel ways of evaluating the impact of PLSs to establish a more evidence-based approach to

PLSs including by conducting reader surveys. While this has been done for graphical abstracts in a social media context, little has been done with text-based PLSs [27,28].

## Conclusion

PLSs have the potential to meaningfully engage people with trustworthy research. Many individuals in health and medical publishing are focused with enthusiasm on the production, publication and distribution of PLSs. Although PLSs have been used in health and medical publishing for several decades [3], ongoing collaboration between all interest holders would help advance this field to the benefit of all involved. Despite barriers, the desire to nudge innovation and adapt PLSs to global trends remains a focus of publishers.

## Supporting information

**S1 Text. Interview guide.**
(DOCX)

## Author contributions

**Conceptualization:** Karen Gainey, Kirsten McCaffery, Danielle Muscat.

**Data curation:** Karen Gainey.

**Formal analysis:** Karen Gainey, Danielle Muscat.

**Investigation:** Karen Gainey.

**Methodology:** Karen Gainey, Kirsten McCaffery, Danielle Muscat.

**Project administration:** Karen Gainey.

**Resources:** Karen Gainey.

**Supervision:** Kirsten McCaffery, Danielle Muscat.

**Validation:** Karen Gainey, Kirsten McCaffery, Danielle Muscat.

**Visualization:** Karen Gainey, Kirsten McCaffery, Danielle Muscat.

**Writing – original draft:** Karen Gainey.

**Writing – review & editing:** Karen Gainey, Kirsten McCaffery, Danielle Muscat.

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
