## [Decision Letter · Decision Letter 0]

31 Jul 2025

PONE-D-25-30641What do editors of medical journals think about opportunities and barriers to advancement in the publication of plain language summaries? A qualitative analysisPLOS ONE?

Dear Dr. Gainey,

We look forward to receiving your revised manuscript.

Kind regards,

Hannah Goss

Academic Editor

PLOS ONE

Journal Requirements:

“I have read the journal's policy and the authors of this manuscript have the following competing interests: Karen M. Gainey, Kirsten J. McCaffery and Danielle M. Muscat have completed the International Committee of Medical Journal Editors (ICMJE) uniform disclosure form at http://www.icmje.org/coi_disclosure.pdf and declare no support from any organisation for the submitted work; no financial relationships with any organisations that might have an interest in the submitted work in the previous 3 years; and no other relationships or activities that could appear to have influenced the submitted work, with the exception of Health Literacy Solutions Pty Ltd, at which Kirsten McCaffrey and Danielle Muscat are directors. Kirsten McCaffery and Danielle Muscat are Editors of health and medical journals.”

3. In the online submission form you indicate that your data is not available for proprietary reasons and have provided a contact point for accessing this data. Please note that your current contact point is a co-author on this manuscript. According to our Data Policy, the contact point must not be an author on the manuscript and must be an institutional contact, ideally not an individual. Please revise your data statement to a non-author institutional point of contact, such as a data access or ethics committee, and send this to us via return email. Please also include contact information for the third party organization, and please include the full citation of where the data can be found.

Additional Editor Comments:

Both reviewers requested some clarification around theoretical perspective and sample size justification.The use of quotes within the results was commended, but would be benefit from greater contextual description of the participants.Both reviewers requested further detail on the implications of findings.

Further line by line comments should also be considered by the authors.

Reviewers' comments:

Reviewer's Responses to Questions

**Comments to the Author**

1. Is the manuscript technically sound, and do the data support the conclusions?

Reviewer #1: Yes

Reviewer #2: Yes

2. Has the statistical analysis been performed appropriately and rigorously?

Reviewer #1: N/A

Reviewer #2: N/A

3. Have the authors made all data underlying the findings in their manuscript fully available?

Reviewer #1: Yes

Reviewer #2: Yes

4. Is the manuscript presented in an intelligible fashion and written in standard English?

Reviewer #1: Yes

Reviewer #2: Yes

Reviewer #1: I believe that this manuscript is worthy of publication, but may some amendment, particularly around providing adequate data on the characteristics of the participants to support interpretation of the data, and also a more through review of the literature surrounding the impact of PLSs than is currently provided.

Please see my comments provided as an attachment.

Reviewer #2: Congratulations to the authors for selecting to investigate such a relevant topic for the current times. It is a complex topic and involves a relatively unreachable participant group, this deserves acknowledgement. I believe the insights from this paper will be a fantastic starting point to dig deeper into systemic challenges that plague the publishing industry, when it comes to offering PLSs.

Here are my recommendations.

Introduction:

Well-written in the context of the current scope of the study. The only suggestion I would have is to assess whether this scoping review can complete the landscape sweep the authors aim to achieve. doi: 10.1007/s40271-024-00700-y

Methods:

I do have a few major questions on this section.

Defining criteria for applying the concept of information power: study aim, sample specificity, established theory, quality of dialogue, analysis strategy

- Do the authors feel that the sample specificity was sufficiently narrow to support using information power rather than saturation?

- Phenomenological analysis or reflexive thematic analysis involves exploration of lived experiences, which is rightly adopted by the authors. However, that doesn't offer the established theory for analysis, that the authors mention. It would be important to understand the rationale for this unique approach.

Data analysis:

- Was any software such as MaxQDA used?

- Was a coding tree developed and used for cross transcript analyses?

- How were disagreements resolved?

Also, I'd recommend using the COREQ COREQ (COnsolidated criteria for REporting Qualitative research) checklist to make the reporting more robust.

Perhaps I missed this but what was KMcC's role in this whole interview and coding process? I'm being mindful of the contributions that qualify for authorship per ICMJE criteria.

Results:

Very well written and embeds the quotes from the participants effectively to support the insights drawn.

Minor:

- 11 to 9 is perhaps not majority for males?

- Was any data collected to understand the exposure of participants to specifically PLS-related decision-making?

- As some participants represented more than one entity, does this make it challenging to analyze their input, as to whether their approach and experiences varied across the journals?

For the thematic analysis reporting, it would be even more impactful if the participants are labeled by which group they're representing, editors or publishers and, whether their journal publishes PLSs.

Curious as to whether these topics came up during to discussions:

- any experiences shared about this perceived "pain" of developing PLSs vs other formats such as visual abstracts?

- whether they worry about an imbalance b/w industry-funded vs academic research PLSs?

- making available or the utility of already available PLS author guides?

- about some publishers developing PLSs themselves?

- how to ensure that PLSs maintain a fair and balanced narrative?

- how to make PLSs more searchable and accessible?

Consumers: Technically an apt term, but in the context of PLSs as these could be patients, general public, etc, could there be a different term such as intended audience, end-user?

Discussion:

Overall, this section can be improved. There is a wealth of insights in the Results section, which deserve more contextualization, to highlight the novelty of this research. For instance, the potential contradiction b/w editors of journals which offer PLSs vs those which don't. This is perhaps even more relevant in the context of perceived barriers.

I also wonder whether the study can be labelled as comprehensive, given the qualitative nature and typical lack of generalizability that comes with it. Detailed for sure, high value no doubt.

Restrictions around language (English only) and no specific consideration to include journals which publicize themselves as patient experience or participatory science journals - both are easy-to-identify limitations. The implications should be discussed under Discussions given that one of the main aims of PLSs is accessibility and inclusiveness.

**Do you want your identity to be public for this peer review?** For information about this choice, including consent withdrawal, please see our Privacy Policy

Reviewer #1: **Yes:** Blair Hesp

Reviewer #2: No

---

## [Author Response · Author response to Decision Letter 1]

16 Sep 2025

Author comments to reviewers – PONE-D-25-30641

REVIEWER ONE

I believe that this manuscript is worthy of publication, but may some amendment, particularly around providing adequate data on the characteristics of the participants to support interpretation of the data, and also a more thorough review of the literature surrounding the impact of PLSs than is currently provided.

This qualitative analysis provides insight into the thoughts and impressions of peer-reviewed journal editors on the adoption of plain-language summaries (PLSs) by medical journals, which is a topic of ongoing debate in terms of how to encourage greater adoption of this innovative format by both journals and authors.

Overall, I think that the manuscript is deserving of publication and offers an opportunity to further contribute to the discussion about the implementation of PLSs. However, there are a few comments that I hope that Authors this manuscript may be able to consider and apply before I believe this manuscript would be fit for publication. For ease, I have simply commented by section rather than splitting my comments in major/minor categories below:

Introduction

Reviewer comment

Would it be possible to mention the heterogeneity of what may be considered a PLS? The term itself has become anything but plain language, given that the likes of the Good Publication Practice Guidelines 2022 offer a very broad definition of just about any derivative of a manuscript, while the lay person would probably consider a PLS to be a version of the abstract written in plain language. I think this is particularly relevant in that the impression I get is that this manuscript intends to focus on written PLSs, excluding video podcasts etc.

Author response

Thank you for your suggestion, which is pertinent given some of the findings from the study regarding non-text based formats. I have added some text to point out the various formats that are broadly considered ‘PLSs’ on Page 3.

“Most commonly, PLSs are text-based; however, many journals offer alternative formats incorporating audio and visual elements, e.g., infographics”

Reviewer comment

Page 10, line 16 – Is the Stoll reference intended to be replaced with a reference number?

Author response

Yes it should have been a reference number. I have corrected this oversight.

Methods

Reviewer comment

Were there any additional limitations on eligibility for this study? In particular, I think it might be pertinent to mention if the subjects could only represent journals publishing in English, given the multinational representation, and this seems to be mentioned in the Discussion on page 40, lines 828–829.

Author response

I agree that this is an important inclusion criteria. We did include this in the Methods section as one of our criteria on Page 6.

“Works or volunteers for a journal that publishes research articles in English”.

Reviewer comment

Table 1, Sample specificity row – Is sample density correct here, when referring to sample specificity? As a casual reader, I would interpret high sample density as suggesting a high proportion of journals who were eligible for this study were sampled.

Author response

I understand your concern. According to Malterud et al (2015), a sample specificity can be considered “dense” when the participants hold information, experience and properties relevant to the research question(s). As such, our sample could be characterized as sufficiently dense. We have added the following in the table for clarification:

“Our sample specificity could be considered dense given that participants belonged to a specified target group while also exhibiting some variation within the experiences to be explored.”

Malterud K, Siersma VD and Guassora AD (2015) Sample Size in Qualitative Interview Studies: Guided by Information Power. Qualitative Health Research. 26(13): 1753-1760. Available from: https://doi.org/10.1177/1049732315617444

Reviewer comment

Table 1, Established theory row – I’m not sure I 100% understand how the application in the current study links to the need for a smaller sample size. I guess I was kind of looking for an explanation of how well established the principle of phenomenology is to justify a smaller sample size rather than an explanation of how it was relevant to this study.

Author response

I have reworded the description ‘Established theory’ in the second column to help with clarity. It now reads “When stronger and more established theories are used, smaller sample size is required.”

This means that, as the theories of phenomenology and reflexive thematic analysis are well established, a smaller sample size is indicated.

Reviewer comment

Table 1, Quality of dialogue row – Are there words missing in the Explanation column or could this be made clearer? It doesn’t feel like it quite makes sense. Is the intention to say that when there is stronger dialogue, a smaller sample size is required?

Author response

Thank you for picking up this oversight. I have added the additional text, which will hopefully provide the necessary clarity.

“When there is stronger dialogue in interviews, a smaller sample size is required.”

Reviewer comment

Table 1, Quality of dialogue row – The second mention of KG as the interviewer appears to be redundant and could be deleted.

Author response

I agree and have made this deletion in Table 1.

Reviewer comment

Page 9, line 158 – This mention of the interview dates also appears to be redundant, given that this is also mentioned on page 7, lines 143-145.

Author response

I have removed the mention of interview dates on page 7.

Reviewer comment

Page 9, line 175 – Is the mention of KG as the interviewer again redundant in light of Table 1, or vice versa?

Author response

I appreciate that there might be some redundancy in this, however, I have kept the mention of KG as interviewer on Page 9 for two reasons. First, some readers may not have read Table 1 thoroughly. Second, it is accepted convention to note the initials of the research team when they are mentioned in the text, for transparency.

Reviewer comment

Page 9, line 176 – I would recommend adding a percentage to the mention of two participants to provide context in terms of the overall study size.

Author response

Thank you for this comment. I have added text at the beginning of this paragraph as well as the percentage of participants (2/20;10%) as suggested, to add clarity to our sample size.

“We conducted 20 semi-structured interviews, guided by the first author (KG)”.

“For the two participants (2/20; 10%) who chose this option, we emailed the raw transcript within two days of their interview.”

Reviewer comment

In the Results, the participants often make reference to past experiences and employment, yet they are being presented as representing their current position. I think it needs to be made clear that the scope of the interviews was not limited to their current position and the participants were free (and encouraged?) to reference past experiences outside of their current position. That diversity of experience is a good thing, but I think this context needs to be given in that the characteristics presented in the results are not 100% reflective of what is being discussed here. Likewise, comments are made about what other people are doing (e.g. page 26, line 353), so this should also be flagged as being within scope.

Author response

I agree with your point and appreciate that this was not included in the Methods section. I have added text to provide clarity to our approach with participants regarding their previous as well as current editorial experience.

“During interviews, participants were encouraged to draw on any previous editorial experience to supplement that from their current position.”

Results

Reviewer comment

Please provide the total number of study participants upfront as part of the participant characteristics.

Author response

I have added this to the Methods section on Page 10, line 172.

Reviewer comment

I recommend providing the exact percentages when mentioning patient characteristics because “most” is an ambiguous term that does not allow the reader to gauge exactly what proportion of the participants fit in any one category.

Author response

I have added frequency and percentages data.

“Most participants were male (11/20; 55%), resided in either the United Kingdom or Australia (13/20; 65%), had worked as an editor for at least five years (15/20; 75%) and reported making up to 10 editorial decisions each week (10/20; 50%). Four (20%) participants were unsure how many editorial decisions they made per week.”

Reviewer comment

Page 12, lines 215¬¬ 216 – The numbers here add up to 24 compared with the N of 20. For clarity, I would suggest presenting these numbers as journal only, publishing group only and both rather than including the both group within each of the other subsets (i.e., counting 2 participants three times).

Author response

I understand how that could be confusing, so I have taken your suggestion. The sentence now reads as “Most (N=17) participants represented a journal and 5 represented a publisher/publishing groups.”

Reviewer comments

Page 12, lines 215¬¬ 216 – Subsets should be presented as lower case n, in line with convention (N being for the overall population) and I would also recommend presenting percentages alongside n numbers for additional context.

Page 13, lines 217–218 – n=11 is described as both half and almost half in the same sentence. Regardless, with such a small sample size, I would generally try and avoid approximations such as “half” because of the small sample size and how the sample was sourced, which means the generalisability of these statements is limited.

Table 3 – N=20 is in the table caption and table. It is only necessary to present this once, perhaps as ‘Characteristics (N=20)’ and then leaving the second column as n (%) (see next comment)?

Table 3 – Rather than two columns, I suggest considering presenting the data as n (%) in one column. This is easier to read and understand compared with two columns with numbers, which initially looks like two arms of a single study.

Author responses

Thank you for your recommendations. I have amended Table 3 accordingly.

Reviewer comments

Table 3, journal publishers – I think this requires greater transparency because the Methods indicate that 5 publishers were targeted, yet an ‘Other’ category is introduced. This may need to be flagged as a possibility in the Methods.

Table 3, journal publishers – The ‘other’ publishers need to be identified. In the age of predatory journal publishers, it is extremely important to name these publishers, possibly in a footnote with n numbers for transparency about who was being interviewed.

Author response

I appreciate the need for transparency. As noted in the Methods, we were open to including journals from publishers other than those that we targeted specifically.

“To engage with editors from journals that publish in the area of health literacy and health communication (given the relevance to PLSs), we included editors from these journals even if they didn’t publish PLSs.”

Regarding the specific journals included in the “Other” category in Table 3, unfortunately authors are unable to provide any journals publishers specifically in this category as that would contravene the terms outlined in the participant consent form, which ensured participant anonymity would be protected. Journals in the “Other” category vary in size, however, all participants and the journals or publishers they represented were investigated prior to interviews to ensure they were not predatory in nature.

Reviewer comment

Table 3, job title – If a n of 1 is provided for Senior Editor and Production Editor, then these should either be bundled in ‘Other’ and all six positions defined in a footnote, or the other positions spelt out, especially if any of them have an n ≥2.

Author response

I appreciate the confusion over the way that the results for job titles have been presented. Unfortunately, more than one job title included in the sub-category of ‘Other’ is so specific that I could easily lead to identification of the participants. I have added a footnote to make this clear. There are no job titles in ‘Other’ with a frequency of >1.

“These job titles have not been included because their specificity could lead to identification of the participants.”

Reviewer comment

Table 3, number of editorial decisions/week – Is the 50+ row necessary if it has a zero value and all the other values add up to 20?

Author response

Although not necessary, the fact that it’s value is zero conveys useful information i.e., there appears to be a cap on the number of perceived editorial decisions per week. For that reason, I have left this row in Table 3.

Reviewer comment

Table 3, general – It would be good to have more granular information about the journals involved, for example, journal scope (e.g., general [N Engl J Med], speciality [Blood], sub-specialty [Pediatr Blood Cancer]; see page 20, line 362 as an example of where ‘specialty’ journals are mentioned and page 46, lines 810–811 where differing journal scope is discussed, without context being provided in the current characteristics table), publication frequency (weekly/monthly etc.), online only vs print, open access only vs pay for access, impact factor (probably categories, e.g., 0–2, 2–4, etc.) to provide context for who the participants are and what sort of journals they are representing because I think that is key to understanding the comments given. Each of these characteristics could have a major impact on decision making around PLSs.

Author response

I appreciate your request for the inclusion of this level of detail. Authors have not included detailed journal information to ensure that the journal in not identifiable by the reader, preserving the anonymity of the participants, as previously discussed.

Reviewer comment

Table 3, employment – Is it possible to outline how many are in paid employment versus voluntary roles. Are the two participants who represent both the publisher and journal publisher employees who at as journal editors, which seems to be becoming more common. That might need to be clarified for context. I apologise if this is somewhat redundant in the context comment 3, but it might help to distinguish academics from paid employees.

Author response

Unfortunately, this data was not collected, so we cannot report it.

Reviewer comment

Quotes – Would it be possible to provide some context for each participant at first mention, such as ‘Participant 3 (Editor-in-Chief, general medical journal)’. I think such detail is important framing for each quote. As noted in the Methods, biases will be present, so I think it is important for the reader to be made aware of some of these potential biases and the context within which these quotes are being made.

Author response

I appreciate that this distinction adds context to the participant examples/quotes. I cannot provide detailed participant information that could be considered identifiable, so I have included information that will add context to the participant examples/quotes without violating participant confidentiality. I have added the following information after each participant example/quote: editor/publisher and journal category. For example “…” Participant X (editor/general medical journal).

To keep this simple, journals have been categorised according to their primary scope. The journal categories are:

• General medical (n=6)

• Specialist medical (n=5)

• Allied health (n=3)

• Public health, health promotion, preventative medicine, healthcare policy (n=4)

• Health & medical education (includes health literacy & health communication journals) (n=5)

I have now added these journal categories to Table 4. Journal characteristics.

Reviewer comment

Page 19, lines 336 – I just want to check that “So, having a like, there’s no for them, maybe that they don’t…” is a correct transcription because it is very broken text and doesn’t seem to have a real flow of thought attached. Perhaps

---

## [Decision Letter · Decision Letter 1]

25 Dec 2025

PONE-D-25-30641R1What do editors of medical journals think about opportunities and barriers to advancement in the publication of plain language summaries? A qualitative analysisPLOS One?

Dear Dr. Gainey,

Thank you for submitting your manuscript to PLOS ONE. After careful consideration, we feel that it has merit but does not fully meet PLOS ONE’s publication criteria as it currently stands. Therefore, we invite you to submit a revised version of the manuscript that addresses the points raised during the review process.

We look forward to receiving your revised manuscript.

Kind regards,

Sadiq H. Abdulhussain, Ph.D.

Academic Editor

PLOS One

Journal Requirements:

Reviewers' comments:

Reviewer's Responses to Questions

**Comments to the Author**

Reviewer #1: All comments have been addressed

Reviewer #3: All comments have been addressed

Reviewer #4: All comments have been addressed

2. Is the manuscript technically sound, and do the data support the conclusions?

Reviewer #1: Yes

Reviewer #3: Yes

Reviewer #4: Yes

3. Has the statistical analysis been performed appropriately and rigorously?

Reviewer #1: N/A

Reviewer #3: Yes

Reviewer #4: N/A

4. Have the authors made all data underlying the findings in their manuscript fully available?

Reviewer #1: Yes

Reviewer #3: Yes

Reviewer #4: Yes

5. Is the manuscript presented in an intelligible fashion and written in standard English?

Reviewer #1: Yes

Reviewer #3: Yes

Reviewer #4: Yes

Reviewer #1: I appreciate the Authors making the effort to incorporate a large number of changes.

While I don't think that the additional information requested would be sufficient to allow the potential identification of the interviewees, I respect that the Authors are in the best position to make that judgement and how the manuscript is presented is their prerogative.

Therefore, I look forward to seeing this published in due course.

Reviewer #3: Thank you for the detailed revisions and point-by-point responses. Overall, I think you have addressed the key concerns from the previous round. In particular, you clarified the heterogeneity of what can be considered a PLS and added text acknowledging non-text formats, which better aligns the scope and findings. You also strengthened methodological transparency by explicitly stating the English-language eligibility criterion, and improved interpretability of participant characteristics by adding frequencies/percentages and correcting presentation issues. The addition of a new table that captures whether PLSs are published or mandatory also responds well to prior suggestions. Lastly, the conflict of interest section appears to have been revised to reflect the requested clarity in disclosure.

Reviewer #4: The comments were attached as a word document with tracked changes. Also, a critical appraisal checklist for qualitative studies was used as a tool during the review.

All pervious comments to the authors were addressed except one. It was added as a comment at the end of the manuscript file just before the references heading.

**Do you want your identity to be public for this peer review?** For information about this choice, including consent withdrawal, please see our Privacy Policy

Reviewer #1: **Yes:** Blair Hesp

Reviewer #3: No

Reviewer #4: No

---

## [Author Response · Author response to Decision Letter 2]

27 Jan 2026

I have provided a detailed response to all reviewer comments in an attached file.

---

## [Editor Report · Decision Letter 2]

1 Feb 2026

What do editors of medical journals think about opportunities and barriers to advancement in the publication of plain language summaries? A qualitative analysis

PONE-D-25-30641R2

Dear Dr. Gainey,

We’re pleased to inform you that your manuscript has been judged scientifically suitable for publication and will be formally accepted for publication once it meets all outstanding technical requirements.

Kind regards,

Sadiq H. Abdulhussain, Ph.D.

Academic Editor

PLOS One

---

## [Editor Report · Acceptance letter]

PONE-D-25-30641R2

PLOS One

Dear Dr. Gainey,

I'm pleased to inform you that your manuscript has been deemed suitable for publication in PLOS One. Congratulations! Your manuscript is now being handed over to our production team.

Kind regards,

on behalf of

Dr. Sadiq H. Abdulhussain

Academic Editor

PLOS One